# The Novel Immune Checkpoint GPR56 Is Expressed on Tumor-Infiltrating Lymphocytes and Selectively Upregulated upon TCR Signaling

**DOI:** 10.3390/cancers14133164

**Published:** 2022-06-28

**Authors:** Vrouyr Bilemjian, Martijn R. Vlaming, Jimena Álvarez Freile, Gerwin Huls, Marco De Bruyn, Edwin Bremer

**Affiliations:** 1Department of Hematology, University Medical Center Groningen, University of Groningen, 9713 GZ Groningen, The Netherlands; v.bilemjian@umcg.nl (V.B.); m.r.vlaming@umcg.nl (M.R.V.); j.alvarez@umcg.nl (J.Á.F.); g.huls@umcg.nl (G.H.); 2Department of Obstetrics and Gynecology, University Medical Center Groningen, University of Groningen, 9713 GZ Groningen, The Netherlands

**Keywords:** cancer immunotherapy, tumor-infiltrated lymphocytes (TIL), GPR56, immune checkpoint

## Abstract

**Simple Summary:**

Despite the clinical efficacy of so-called immune checkpoint inhibitors (ICIs) in various cancers, some cancer types, including epithelial ovarian cancer (EOC), do not effectively respond to current therapeutics. Thus, the identification of new immune checkpoints that regulate T cell immunity remains of great interest. One as yet largely uninvestigated checkpoint of potential interest is the G protein-coupled receptor 56 (GPR56), which belongs to the adhesion GPCR family. We identified that GPR56 is expressed on tumor infiltrating lymphocytes (TILs) and investigated its role as a potential immune checkpoint within the context of cancer. Based on our data, GPR56 indeed appears to function as an immune checkpoint in TILs and may thus provide a novel immunotherapeutic target for the reactivation of tumor-infiltrating and tumor-reactive lymphocytes.

**Abstract:**

High levels of tumor-infiltrating lymphocytes (TILs) in the tumor microenvironment (TME) are associated with a survival benefit in various cancer types and the targeted (re)activation of TILs is an attractive therapeutic anti-cancer approach that yields curative responses. However, current T cell targeting strategies directed at known immune checkpoints have not increased objective response rates for all cancer types, including for epithelial ovarian cancer (EOC). For this reason, the identification of new immune checkpoints that regulate T cell immunity remains of great interest. One yet largely uninvestigated checkpoint of potential interest is the G protein-coupled receptor 56 (GPR56), which belongs to the adhesion GPCR family. GPR56 was originally reported to function in cerebral cortical development and in anti-depressant response, but also in cancer. Recently, GPR56 was identified as an inhibitory receptor expressed on human NK cells that by cis-interaction with the tetraspanin CD81 attenuated the cytotoxic activity of NK cells. This NK cell checkpoint could be blocked by an GPR56 antibody, leading to increased cytotoxicity. Interestingly, GPR56 expression has also been reported on cytokine producing memory CD8 T lymphocytes and may thus represent a T cell checkpoint as well. Here, GPR56 mRNA expression was characterized in the context of TILs, with GPR56 expression being detected predominantly in tumor infiltrating CD8 T cells with a cytotoxic and (pre-)exhausted phenotype. In accordance with this mRNA profile, TILs from ovarian cancer patients expressed GPR56 primarily within the effector memory and central memory T cell subsets. On T cells from healthy donors the expression was limited to effector memory and terminally differentiated T cells. Notably, GPR56 expression further increased on TILs upon T cell receptor (TCR)-mediated stimulation in co-cultures with cancer cells, whereas GPR56 expression on healthy primary human T cells did not. Further, the ectopic expression of GPR56 significantly reduced the migration of GPR56-positive T cells. Taken together, GPR56 is a potential immune-checkpoint in EOC found on (pre-)exhausted CD8 TILs that may regulate migratory behavior.

## 1. Introduction

Within the tumor microenvironment (TME), a variety of immuno-suppressive mechanisms serve to shut-down anti-tumor T cell immunity. These range from the presence of inhibitory immune cell subtypes, inhibitory cytokines, and chemokines to metabolic competition and the expression of inhibitory receptors, such as PD-L1 and CTLA-4. Collectively, these shape the activity and cytotoxic potential of the tumor infiltrating immune repertoire, yielding distinct immune landscapes across different cancer types and within individual patients [1,2,3]. In the past decades, diverse therapeutic approaches have been developed to specifically counteract these inhibitory signals, with the development of immune checkpoint inhibitors (ICIs) reaching a revolutionary milestone in the field of immuno-oncology. Most notably, ICIs targeting CTLA-4 or PD-L1/PD-1 have yielded remarkable clinical responses for a variety of cancer types, with a range of drugs currently approved targeting these axes available as cancer treatments (reviewed in [4]). 

However, despite the introduction of these promising ICIs some cancer types do not effectively respond. For instance, the treatment of PD-1 antibody Nivolumab, PD-L1 antibody Avelumab, PD-1 antibody Pembrolizumab, and CTLA-4 blocking antibody MDX-CTLA4 only yielded objective response rates (ORR) up to 15% in epithelial ovarian cancer (EOC) [5,6,7,8]. Similarly, the treatment of breast cancer and pancreatic cancer still face challenges, despite the introduction of ICIs (as reviewed in [9,10]). To improve the therapeutic outcome for ovarian cancer and other advanced malignancies, novel combinations of blocking antibodies targeting PD-L1, PD-1, and CTLA-4, with or without chemotherapeutics or immune-stimulating antibodies targeting 4-1BB or OX-40, are currently being clinically evaluated (ClinicalTrials.gov Identifier: NCT02608684, NCT03249142, NCT02554812). 

Besides focusing on novel therapeutic combinations of known targets, various new immune-checkpoints have been identified, including LAG-3 [11], TIM-3 [12], TIGIT [13], VISTA [14], HLA-G [15], NKG2A [16], HHLA2 [17], CD38 [18], and HVEM [19]. One particular, less established immune-checkpoint is the protein GPR56. GPR56 is an adhesion G-protein-coupled receptor with an array of functions, ranging from cortical development [20,21], anti-depressant response [22], hematopoietic development [23], to tumor cell adhesion and progression [24,25,26]. High GPR56 expression on tumor cells associated with a decreased 5-year overall survival rate in colorectal cancer and acute myeloid leukemia [27,28]. Further, on NK cells GPR56 acts as an inhibitory receptor through cis-interaction with the tetraspanin molecule CD81 [29,30]. Driven by transcription factor Hobit, especially fully differentiated NK cells express GPR56, which limits effector activity. Upon activation, Hobit was rapidly downregulated, thereby downregulating GPR56 and unleashing NK effector activity [29]. Interestingly, GPR56 expression has also been reported on a subset of cytotoxic T lymphocytes and on cytokine producing memory T cell subsets, on which it regulates chemotaxis [31,32,33].

GPR56 expression and function have not been evaluated in the context of tumor infiltrating lymphocytes (TILs) yet. As the magnitude, composition, quality, and phenotypic features of the TIL population have been linked to treatment response-rates and outcome (as reviewed in [34], we here set out to identify GPR56 expression on TIL fractions from various tumor types and to subsequently phenotypically define these populations by consulting RNA sequencing datasets. Cytotoxic and terminally exhausted TIL subsets displayed cytotoxic, (pre-)exhausted and tumor-reactive signatures. Further, GPR56 expression levels, as well as expression of several effector/co-stimulatory molecules identified in the sequencing analysis, were confirmed on TILs from epithelial ovarian cancer (EOC) patients. Interestingly, TILs upregulated GPR56 expression upon TCR stimulation ex vivo, whereas primary healthy T cells did not. Furthermore, the ectopic expression of GPR56 in T cells reduced migratory potential in response to chemoattractants and T cell activation in a T cell: tumor cell co-culture reporter assay. Together, GPR56 may be a potential novel immunotherapeutic target for reactivation of tumor-infiltrating and tumor-reactive lymphocytes.

## 2. Materials and Methods

### 2.1. Analysis of Single Cell mRNA Sequencing Data

Raw sequencing counts and metadata from the Gene Expression Omnibus (GEO) dataset GSE99254 was downloaded (T cell landscape of non-small cell lung cancer revealed by deep single-cell RNA sequencing), consisting of 12,346 T cell samples isolated from the blood, healthy adjacent tissue and tumor tissue of 14 treatment-naïve non-small-cell lung cancer (NSCLC) patients [35]. The data were ingested into Seurat V4 in R language version 4.0.3. The data were log normalized (10,000 as the scale factor) and scaled. Further clustering was done using first 30 PCs based on the elbow plot distribution of PCs. Clusters were annotated based on the expression of CD8A, CD4, and FOXP3. The cluster with double-positive expression of CD8A and CD4 was discarded. Most of the cells were negative for ADGRG1 (GPR56), with cells with non-zero GPR56 expression being classified as GPR56-positive. Differential expression was performed in Seurat using FindMarkers function with MAST as the method of choice.

The dataset from the tumor Immune Cell Atlas study [36] was downloaded in the form of an RDS file containing the Seurat object. The cell types in the study were already defined. The following cell types were included in our study: regulatory T cells, T helper cells, Th17 cells, recently activated CD4 T cells, naive-memory CD4 T cells, Transitional memory CD4 T cells, naive T cells, proliferative T cells, pre-exhausted CD8 T cells, cytotoxic CD8 T cells, effector memory CD8 T cells, and terminally exhausted CD8 T cells. Most of the cells were negative for GPR56, with cells having non-zero GPR56 expression being considered GPR56-positive. Differential expression was calculated by using FindMarkers function from Seurat with MAST as the method of choice [37].

### 2.2. Isolation of Peripheral Blood Mononuclear Cells (PBMCs)

Buffy coats were purchased from Sanquin and all donors provided informed consent (Sanquin Blood Supply, Groningen, The Netherlands). Human PBMCs were isolated via Lymphoprep density gradient centrifugation. T cells were isolated using an autoMACS Pro Separator (Miltenyi Biotec, Bergisch Gladbach, North Rhine-Westphalia, Germany) and a Pan T Cell Isolation Kit (Miltenyi Biotec, Bergisch Gladbach, North Rhine-Westphalia, Germany) following the manufacturer’s recommendations. After isolation, T cells were suspended in RPMI with 10% FCS.

### 2.3. Isolation of Tumor Infiltrated Lymphocytes (TILs) from Fresh Tumor Tissue

Tumor infiltrating lymphocyte extraction was performed on ovarian cancer tissue obtained during surgery collected in the University Medical Center Groningen, The Netherlands. This study was carried out in the Netherlands in accordance with International Ethical and Professional Guidelines (the Declaration of Helsinki and the International Conference on Harmonization Guidelines for Good Clinical Practice. The use of anonymous waste material is regulated under the code for good clinical practice in the Netherlands [38]. Patients had given consent for the use of surgical material for research purposes. TILs used for analyses were isolated from fresh tumor samples obtained during cytoreductive surgery. Small tumor pieces of approximately 0.5 cm^3^ were cut with scalpel and digested in digestion medium (RPMI supplemented with 1 mg/mL collagenase type IV (Life Technologies, Carlsbad, CA, USA) and 31 U/mL rhDNase (Pulmozyme, Genentech, South San Francisco, CA, USA)) for 30 min at 37 °C. Subsequently, the digestion medium containing remaining tumor pieces and TILs was filtered using a 70-μm cell strainer (Corning, Corning, NY, USA), whereupon cells were pelleted, washed, and cryopreserved until further use. 

### 2.4. Multiparameter Flow Cytometry

TILs from the digested tumor samples were phenotyped by multiparameter flow cytometry (CytoFLEX Flow Cytometer, Beckman Coulter, Brea, CA, USA). Zombie Aqua Fixable Viability Kit was used for live/dead stain according to the manufacturer’s instructions (BioLegend, San Diego, CA, USA). Antibodies used for phenotyping were CD3-PerCP-Cy5.5 (OKT3), CD8-APC-eFluor780 (RPA-T8), CD4-PE, and CD45RO-PE-Cy7 (UCHL1) (all from eBioscience, San Diego, CA, USA); CCR7-BV421 (150503; BD Biosciences, San Jose, CA, USA), GPR56-FITC (2446B; R & D systems, Minneapolis, MN, USA) CD103-APC (Ber-ACT8; BD Biosciences), CD25-APC (MEM-181) and CD69-APC (IT8G1) (both from Immunotools, Gladiolenweg, Friesoythe, Germany). The samples were analysed by using the accompanying CytExpert software v2.4.

### 2.5. Cell Lines and Transfectants

Cell line ES-2 was obtained from the American Type Culture Collection (ATCC). Cells were cultured in RPMI-1640 or DMEM (Lonza, Basel, Switzerland), supplemented with 10% fetal calf serum (FCS, Thermo Scientific, Waltham, MA, USA). The artificial CD3 scFv-presenting cell lines MDA-MB231scFvCD3, ES-2scFvCD3, FaDuscFvCD3, and DLD1scFvCD3 are based on the Lentiviral synNotch receptor construct pHR_PGK_antiCD19_synNotch_Gal4VP64, which was a gift from Wendell Lim (Addgene plasmid # 79125). The antiCD19 scFv was replaced for CD3 antibody fragment UCHT-1v9 using Gibson cloning, yielding pHR_PGK_scFvCD3_synNotch_Gal4VP64. Lentivirus was produced by transient transfection of HEK293 T cells with psPAX2 and VSV-G packing system using FuGENE (Promega, Madison, WI, USA). Viral supernatants were collected and filtered through a 0.2-μm filter (Eppendorf, Hamburg, Germany). Transduction was performed by adding 1.5 mL viral supernatant to 1.5 mL of RPMI medium (Lonza) containing 0.250 × 106 pre-seeded cells in a 6 well tissue culture plate (Corning) in the presence of 4 μg/mL polybrene (Sigma-Aldrich, St. Louis, MO, USA). Transduced cells were sorted for expression of a Myc-tag (fused to the extracellular CD3 scFv) using anti-Myc mAb Alexa Fluor 647 (clone 9B11, Cell Signaling, Danvers, MA, USA) with a Sony cell sorter sh800s. GPR56 overexpressing cell lines were generated by transducing the cells with lentivirus containing pRRL-SFFV-GPR56-iGFP. After 3 days the transduction efficiency was evaluated by flow cytometry. 

### 2.6. T Cell and TIL Activation Assay

A total of 2000 ES-2scFvCD3 or FaDuscFvCD3 were plated per a well in their respective growth medium. After 24 h, a total of 2 × 10^4^ T cells (Pan T Cell Isolation Kit was used to isolate CD3+ T cells from human PBMC by using autoMACS^®^ Pro Separator, Miltenyi biotec, Bergisch Gladbach, North Rhine-Westphalia, Germany) or TILs (from the digested tumor samples) were added to each well. After 4 days of co-culture GPR56 together with the rest of the T cell markers were analysed by flow cytometry (CytoFLEX Flow Cytometer, Beckman Coulter, Brea, CA, USA).

### 2.7. In Vitro Migration Assay

SDF-1 (CXCL12, Immunotools GmbH, Friesoythe, Germany) was added as chemoattractant to the lower compartment of transwell plates (Corning, Corning, NY, USA) at 0–500 ng/mL in 200 µL of migration buffer (serum-free RPMI, 100 U/mL rhIL-2, 0.3% HSA). Then, primary healthy T cells lentivirally transduced with pRRL-SFFV-GPR56-iGFP or empty vector were loaded in 100 µL RPMI on transwell filters with a pore size of 5 µm (Corning). After 2.5 h of incubation at 37 °C, cells in the lower compartment were harvested, mixed with 30 µL of counting beads (Miltenyi Biotec, Bergisch Gladbach, Germany), and quantified by flow cytometry at a fixed volume of 40 µL and at high speed. The percentage of migration was calculated considering the migration of the primary healthy T cells transduced with the empty vector to be 100%. A two-tailed unpaired *t* test was used to compare the levels of migration between the two groups.

### 2.8. Statistical Analysis

Data are presented as means ± SD as stated in the figure legends. Statistical significance was determined as indicated in the figure legends, with *p* < 0.05 considered statistically significant.

## 3. Results

### 3.1. GPR56 Is Expressed in Cytotoxic and Terminally Exhausted CD8 TILs That Display a Tumor-Reactive Phenotype across Multiple Tumor Types

Within a single-cell tumor immune transcriptomic (scRNAseq) dataset covering different tumor types, GPR56 mRNA expression was detected in various tumor-infiltrating immune cell sub-types, specifically in regulatory T cells, T helper cells, Th 17 cells, proliferative T cells, cytotoxic CD8 T cells, terminally exhausted CD8 T cells and NK cells (Figure 1A,B). Little to no GPR56 expression was observed in B cells, dendritic cells, mast cells and monocytes/macrophages, although some GPR56 expression was detected in M2 tumor-associated macrophages (TAMs). Within the different T cell subsets, GPR56 expression was detected within the cytotoxic and terminally exhausted CD8 T cells subsets (Figure 1B, turquoise and light blue; see Appendix A for mean values in blue bars). As GPR56 expression on NK cells was previously described to attenuate the cytotoxic activity of NK cells by cis-interaction with CD81, its expression was also evaluated in the same data set (Appendix A). We observed CD81 expression on different infiltrated immune subsets within the tumor, including cytotoxic and terminally exhausted CD8 T cells that expressed GPR56.

Analysis of the differentially expressed genes (DEGs) of the GPR56-positive and GPR56-negative cytotoxic CD8 T cells, terminally exhausted CD8 T cells, TH17 cells, proliferative T cells, regulatory T cells, and T helper cells revealed that, across multiple immune cell subtypes, the GPR56-positive population presented significantly higher expression of genes associated with cytotoxic/(pre-)exhausted/tumor-reactive gene signatures (Figure 1C, highlighted in orange). For example, several genes belonging to the tumor necrosis factor receptor superfamily (TNFRSF) were upregulated (e.g., TNFRSF18 (GITR), TNFRSF9 (4-1BB), TNFRSF4 (OX40), TNFRSF1B) across different immune cell sub-types (Figure 1C, excluding the regulatory T cells). Further, genes associated with an enhanced effector state (e.g., ADGRE5, GNLY, CD82, CD7 and several granzymes) as well as genes associated with T cell (pre-)exhaustion and tumor-reactive features, such as HAVCR2, LAG3, PDCD1, TIGIT, and ENTPD1 (CD39), were significantly higher in GPR56-positive T cells. In addition, GPR56-positive cells expressed higher mRNA levels of CXCL13 and TGF-β1, two coordinating players in the anti-tumor immune response. Collectively, the scRNAseq analysis revealed that GPR56-positive TILs display an (pre-)exhausted and tumor-reactive phenotype.

### 3.2. Compared to Blood and Adjacent Tissue, Tumor Localized GPR56-Positive CD8 T Cells Display a Distinct Gene Expression Profile with Upregulated Tumor-Reactive Markers

A second scRNAseq dataset, based on T cell samples isolated from the blood, tumor tissue, and healthy adjacent tissue from treatment-naïve NSCLC patients was used to further evaluate GPR56 mRNA expression in the TIL repertoire. This analysis revealed that, with samples from the three anatomical T cell locations combined, GPR56 expression was primarily detected within the CD8 T cell fraction (Figure 2A, overlapping CD8 and GPR56 purple area’s). Compared to GPR56 expression in CD8 T cells, relatively low expression levels were detected within the CD4 T cell fraction, and no expression of GPR56 was detected within the FOXP3-expressing population (Figure 2A, CD4 and FOXP3 area’s vs. GPR56). Interestingly, whereas for some patients relatively high GPR56 expression was detected, others displayed little to no GPR56 expression, highlighting a clear inter-patient variability for GPR56 expression (Figure 2B). When the GPR56-expressing cells were subsequently subdivided into their anatomical isolation sites, GPR56-expressing T cell fractions were identified in all three anatomical sites, with detected GPR56 expression levels being slightly higher within the blood and healthy adjacent tissue fractions compared to the expression levels found in cells isolated from tumor tissue (Figure 2C).

Evaluating the DEGs within the tumor CD8 GPR56-positive vs. the GPR56-negative T cell fractions, again revealed cytotoxic/(pre-)exhausted/tumor-reactive signatures, reflected by the upregulation of genes, such as ENTPD1 (CD39), ITGAE (CD103), CXCL13, TNFRSF9 (4-1BB), HAVCR2, CTLA4, and CX3CR1 (Figure 2D, highlighted in orange). Interestingly, comparing DEGs within the GPR56-positive fractions isolated from the blood, healthy adjacent tissue and tumor tissue revealed that especially the GPR56-positive CD8 T cells isolated from the tumor tissue were strongly associated with a cytotoxic/(pre-)exhausted/tumor-reactive signature. Specifically, compared to the GPR56-positive vs. the GRP56-negative analysis within the tumor tissue (Figure 2D, left), more T cell effector, co-stimulatory and migratory molecules were detected in the tumor GPR56-positive vs.. blood GPR56-positive or adjacent tissue GPR56-positive analysis (e.g., CD27, CXCR3, TNFSF10 (TRAIL), CD69, ZNF683 (Hobit)) (Figure 2D, middle, right, highlighted in orange). Further, tumor GPR56-positive T cells presented a higher frequency of positivity for proliferation marker Ki-67 compared to GPR56-positive T cells from the blood, suggesting the presence of proliferating T cells within the tumor (Figure 2D, middle). Thus, GPR56-positive T cells reside within the blood, tumor tissue and healthy adjacent tissue, with especially GPR56-positive TILs displaying an (pre-)exhausted and tumor-reactive phenotype.

With both the Pan Cancer and the NSCLC datasets, a regression analysis was performed to assess the correlation of GPR56 with other genes (Appendix A). This revealed correlations with several (pre-)exhaustion genes, such as TIM3 and LAG3, as well as genes involved in T cell effector, co-stimulatory, and migratory functions, such as adgre5, GZMB, and GNLY.

### 3.3. GPR56 Expression Is Elevated on Ovarian Cancer TILs Compared to Healthy Peripheral Blood Lymphocytes and Associates with an Effector Memory Phenotype

GPR56 was subsequently evaluated on healthy primary human T cells and TILs isolated from ovarian cancer patients. This revealed higher expression levels of GPR56 on the total CD3 population (4% vs. 15%) as well as on the CD4 (1% vs. 15%) and CD8 (8% vs. 11%) fractions of TILs compared to healthy primary human T cells, although clear patient variability was observed (Figure 3A,B, red vs. grey bars) (see Appendix A for gating strategy). GPR56-expressing TILs were predominantly identified within the effector memory (TEM) and central memory (TCM) T cell subsets, both on CD4 TILs (Figure 3C: 51% TEM, 28% TCM) and CD8 TILs (Figure 3C: 74% TEM, 19% TCM). On healthy T cells, GPR56 expression was limited to effector memory and terminally differentiated T cells (TEMRA), both on CD4 PBLs (Figure 3C: 52% TEM, 38% TEMRA) and CD8 PBLs (Figure 3C: 29% TEM, 64% TEMRA). Further, when distinguishing between tissue resident (CD103+) and stromal T cells (CD103−), no significant difference was observed in GPR56 expression (Appendix A). 

Importantly, the effector and co-stimulatory molecules CD69, CD25, OX40, and 4-1BB that were identified in the scRNAseq analysis also proved to be upregulated at the protein level on GPR56-positive TILs from three different ovarian cancer donors (Figure 3D). In conclusion, the in vitro evaluation of healthy primary human T cells and TILs isolated form ovarian cancer patients confirmed GPR56 expression on both populations, with the latter displaying the highest expression levels, combined with a predominantly TEM and TCM phenotype that expressed key proteins identified from the scRNAseq analysis.

### 3.4. TCR Stimulation Upregulates GPR56 Expression on TILs and Enhances the Activation Status of T Cells

Upon MHC-independent activation of TCR signaling by anti-CD3 single-chain antibody fragment-expressing ES-2 cells (ES-2scFvCD3), GPR56-expressing TILs upregulated GPR56 expression by approximately three-fold (CD3, 5% to 15%; CD4, 6% to 16%; CD8, 3% to 12%) (Figure 4A, red vs. grey bars). Such an increase was not detected on TCR-stimulated healthy primary human T cells. Before TCR signaling was initiated, comparable GPR56 expression level ratios were found on healthy primary human T cells and TILs, as observed earlier and displayed in Figure 3B (Figure 4A, CD3, 1% vs. 5%, CD4, 1% vs. 6%, CD8, 3% vs. 3%). 

MHC-independent activation of TCR signaling was also subsequently induced by ES-2scFvCD3 or FADUscFvCD3 cells in T cell line Jurkat ectopically expressing GPR56 (Jurkat.GPR56). This revealed a significantly stronger upregulation of activation markers CD25 and CD69 on Jurkat.GPR56 compared to wild type Jurkat (Jurkat.WT) or control Jurkat cells (Jurkat.EV) (Figure 4B). Specifically, after 24 h of TCR stimulation by ES-2scFvCD3 or FADUscFvCD3, Jurkat.GPR56 cells displayed, respectively, 15.5% and 16.5% higher CD69 expression compared to TCR stimulated Jurkat.WT and Jurkat.EV cells (Figure 4B, left graphs, 24 h, grey bars vs. blue and red). 

Concerning CD25 expression, after 24 h of TCR stimulation by ES-2scFvCD3 or FADUscFvCD3 cells, Jurkat.GPR56 cells displayed, respectively, 9.5% and 11% higher CD25 expression compared to TCR stimulated Jurkat.WT and Jurkat.EV cells (Figure 4B, right graphs, 24 h, grey bars vs. blue and red). Together, TILs upregulated GPR56 expression upon TCR stimulation and that TCR stimulated GPR56 expressing T cells feature enhanced expression of activation markers CD25 and CD69. This suggests a link between GPR56 expression and the activation status of T cells. 

As GPR56 NK cell expression was previously described to attenuate the cytotoxic activity of NK cells by cis-interaction with CD81, its expression on CD3-positive Jurkat GPR56 cells was evaluated with Jurkat.GPR56 cells also expressing surface CD81 (Figure 3C). Upon the subsequent co-culture of GPR56-expressing Jurkat.NFAT-luc reporter cells with ES2-anti-CD3 cells, both GPR56-expressing and control Jurkat.NFAT-luc cells increased luminescence levels compared to the Jurkat.NFAT-luc cells alone (Figure 4D, Jurkat only vs. +ES2-anti-CD3). However, a significant decrease in luminescence was detected for Jurkat.NFAT-luc cells expressing GPR56 compared to control cells (Figure 4D, left, light vs. dark grey). Thus, GPR56 inhibited T cell activation in the Jurkat.NFAT-luc model system.

### 3.5. Ectopic GPR56 Expression Inhibits Migration of Primary Healthy T Cells

To assess the possible regulatory effects of GPR56 in T cells, the migration potential of ectopically GPR56-expressing T cells in response to chemoattractant SDF-1 (CXCL12) was examined in a trans-well system. As shown in Figure 5A, ectopic GPR56 expression was successfully introduced on primary healthy T cells. A reduced migratory potential towards chemoattractant SDF-1 (CXCL12) was subsequently observed for primary healthy T cells ectopically expressing GPR56 compared to empty vector-transduced primary healthy T cells (Figure 5B, grey vs. red dots). Thus, GPR56 expression on primary healthy T cells affects in vitro migration in response to chemoattractant SDF-1.

## 4. Discussion

In the current study, we identified that GPR56 mRNA expression was mostly detected in cytotoxic and/or (pre-)exhausted CD8 TILs. GPR56 expression correlated with the upregulation of genes associated with cytotoxic and (pre-)exhausted gene signatures, together often linked to tumor-reactive lymphocyte fractions [39,40,41]. As the exhausted gene signatures in GPR56 expressing TILs were accompanied by the expression of genes, such as IFNg, Ki67, and granzymes, the term (pre-)exhausted was introduced for these cells, encompassing both phenotypes. Comparable signatures have been attributed to (pre-)exhausted TILs before [35,42]. Further, it was demonstrated that GPR56-expressing CD8 T cells isolated from the tumor tissue also display tumor-reactive gene signatures compared to GPR56-positive CD8 T cells isolated from the blood or healthy tumor-adjacent tissue. The expression levels of GPR56, the accompanied effector memory and central memory T cell phenotypes, and the expression of several cell-surface effector molecules were subsequently confirmed in GPR56-positive of ovarian cancer TILs. Furthermore, TCR stimulation in ovarian cancer TILs enhanced GPR56 expression and T cells ectopically expressing GPR56, displaying the enhanced expression of the activation and co-stimulatory markers upon TCR stimulation. A reduced migratory potential towards chemoattractant SDF-1 was also observed for T cells ectopically expressing GRP56, whereas for T cell activation in a T cell, tumor cell co-culture reporter assay was also reduced in T cells expressing GPR56.

The potential immune-checkpoint candidate GPR56 was, in the current study, further evaluated in the context of TILs. Although GPR56-expression, driven by transcription factor Hobit, inhibits the natural cytotoxicity of human NK cells by cis-interaction with the tetraspanin CD81 [29], it appears that GPR56-expression in CD8 TILs cells defines a cytotoxic and tumor-reactive population. Interestingly, transcription factor Hobit was also found to be upregulated in GPR56-positive CD8 T cells isolated from tumor tissue vs. GPR56-positive CD8 T cells isolated from the blood or from normal adjacent tissue. The upregulated expression of transcriptional regulator Hobit was also previously observed in GPR56-positive CD4 TEMRA cells, with these cells also being more clonally expanded compared to the GPR56-negative fraction [33,43,44,45]. In the current report, within PBL GPR56-positve fractions, a large portion of the cells featured a TEMRA phenotype, whereas in GPR56-positive TILs, a shift towards a TEM phenotype was observed. Besides CD4-positive TEMRA cells, GPR56 was also previously identified within a core transcriptome signature of human memory CX3CR1-positive CD8 T cells [46], and within a gene signature of CD39 and CD103 double-positive tumor-reactive CD8 T cells [39]. These signatures were also found in the current study across multiple tumor types. The findings in this study support the suggestion that GPR56-positive T cells represent a more clonally expanded cell population [43] as well as our findings that especially GPR56-positive T cells isolated from tumor tissue display this distinct gene expression profile with upregulated tumor-reactive markers.

Although limited to ovarian cancer TILs and healthy primary human T cells, our in vitro evaluations confirmed GPR56-expression and several accompanied co-stimulatory/activation markers on these cell populations. Using a model system that enabled MHC-independent activation of TCR signaling by anti-CD3 single-chain antibody fragment-expressing tumor cell lines, it was demonstrated that only TILs, and not PBLs, upregulate GPR56 expression upon TCR stimulation. Although follow-up studies are required to characterize the TCR repertoire diversity of these populations, they might reflect a more clonally expanding T cell population. Further, the enhanced expression of activation markers CD25 and CD69 on GPR56-expressing T cell line Jurkat was found upon TCR signaling, strengthening the suggestion that T cell GPR56 expression potentiates their activation status upon TCR stimulation. It would be interesting to assess a possible further enhanced activation status upon treatment with a GPR56 blocking mAb. This evaluation could be substantiated with analysis of pro-inflammatory cytokine production in future studies, strengthening the findings. 

Interestingly, upon evaluation of the migratory potential of primary healthy T cells ectopically expressing GPR56, reduced migration was observed for T cells ectopically expressing GPR56 in response to chemoattractant SDF-1. This observation matches previous studies in which GPR56 was shown to inhibit chemotaxis of human Natural Killer and TEMRA lymphocytes [31,32]. GPR56 has also been held responsible for its negative impact on the migratory potential of neural progenitor cells [47,48] and has been linked to the metastasis of melanoma lesions [49]. Interestingly, the negative impact on the migratory potential of neural progenitor cells was induced by an agonistic anti-GPR56 mAb. Another GPR56-mAb has been shown to potentiate Src–Fak-based adhesion signaling, increasing adhesion to collagen by cells ectopically expressing GPR56 [50]. In the context of cytotoxicity, GPR56 was previously identified as an inhibitory receptor expressed on human NK cells, which, by cis-interaction with the tetraspanin CD81, attenuated the cytotoxic activity of NK cells [29]. This NK cell checkpoint could be blocked by a GPR56 mAb crosslinking GPR56 and, thereby, cause a dissociation of the GPR56-CD81 complex, leading to increased cytotoxicity. In the current study, however, enhanced GPR56 expression was not associated with enhanced CD81 expression, likely suggesting that other GPR56 interacting complexes are present on T cells. Consistent with data from the current study, it is suggested that GPR56 could function as an inhibitory T cell checkpoint. Hence, the next step remains to evaluate the impact of a GPR56 agonist on the cytotoxic potential and migratory capacity of GPR56-expressing primary human T cells.

Together, GPR56 expression in TILs reflects a cytotoxic and (pre-)exhausted T cell fraction, which could be of potential interest for targeted reactivation by immunotherapeutic strategies to release their tumor-reactive cytotoxic potential.

## 5. Conclusions

In the current study, we identified that the G-protein Coupled Receptor (GPR56) appears to function as an immune-checkpoint that may regulate activity of Tumor Infiltrated Lymphocytes. Thus, GPR56 may be a novel target for cancer immunotherapy.

## Figures and Tables

**Figure 1 cancers-14-03164-f001:**
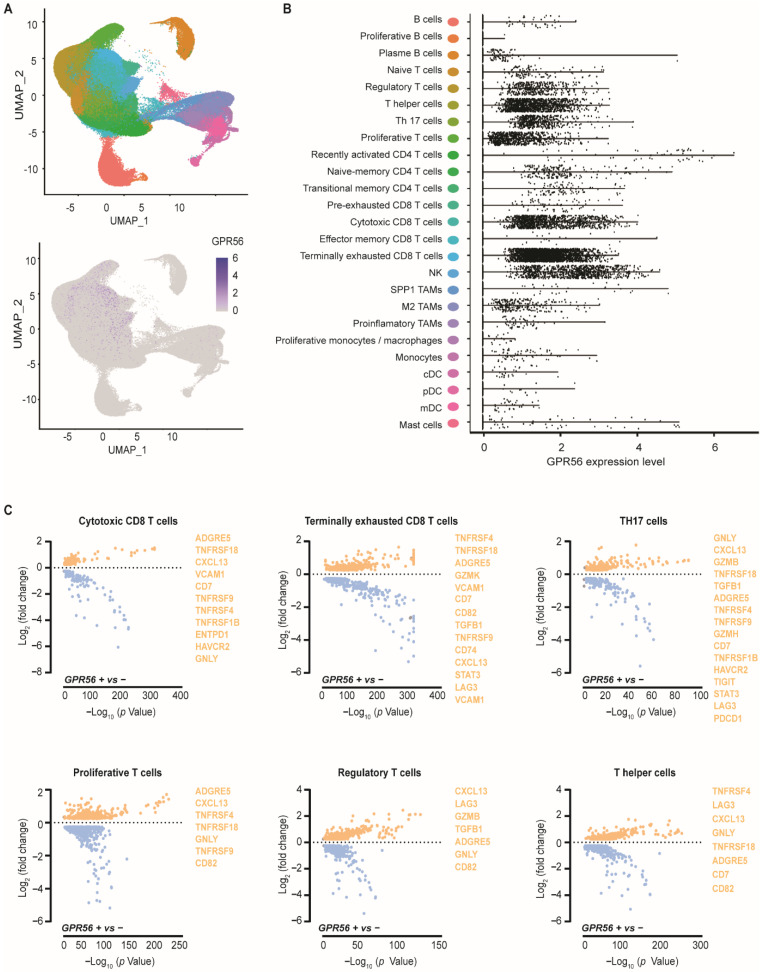
GPR56 is expressed in cytotoxic and terminally exhausted CD8 TILs that display a tumor-reactive phenotype across multiple tumor types. (**A**) Single-cell tumor immune atlas RNA sequencing data-set based on over 500,000 cells from 217 patients and 13 cancer types revealing GPR56 expression within different immune cell subtypes. Visualized by a UMAP representation and in a grouped analysis. (**B**) displaying the different GPR56 expressing tumor infiltrating immune-cell subtypes. (**C**) Differential gene expression of the GPR56-positive vs. GPR56-negative populations within the cytotoxic CD8 T cells, terminally exhausted CD8 T cells, TH17 cells, proliferative T cells, regulatory T cells and T helper cells fractions, calculated using FindMarkers function from Seurat with MAST as the method of choice. Genes with a *p*-value ≤ 0.05 and log2FoldChange ≥ 0.25 are displayed, visualized in volcano plots using GraphPad Prism 8. Several upregulated effector, (pre-)exhausted and tumor reactive genes are highlighted in orange.

**Figure 2 cancers-14-03164-f002:**
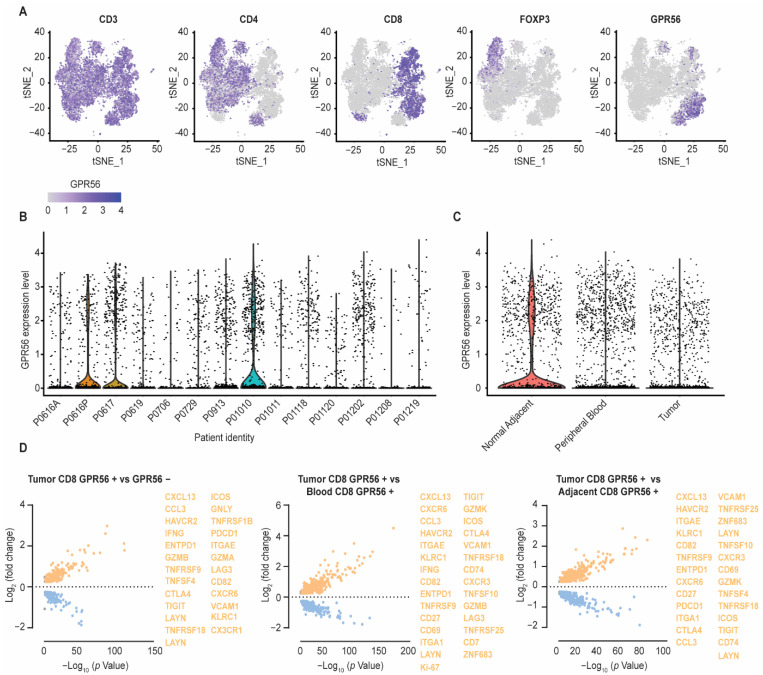
Compared to blood and adjacent tissue, tumor localized GPR56-positive CD8 T cells display a distinct gene expression profile with upregulated tumor-reactive markers. (**A**) GPR56 expression visualized within the CD3, CD4, CD8 and FOXP3-expresssing cell subsets by dimensional reduction analysis (t-SNE) using a single-cell RNA sequencing data set, consisting of 12,346 T cell samples isolated from the blood, healthy adjacent tissue and tumor tissue of 14 treatment-naïve non-small-cell lung cancer (NSCLC) patients. (**B**) Inter-patient GPR56 expression variability. (**C**) GPR56 expressing cells subdivided into their anatomical isolation sites (blood, healthy adjacent tissue and tumor tissue). (**D**) Differential gene expression of the GPR56-positive vs. GPR56-negative population within the CD8 T cell fraction isolated from tumor tissue (left) and differential gene expression of the GPR56-positive CD8 T cell fraction isolated from the tumor tissue vs. the GPR56-positive CD8 T cell fraction isolated from the blood (middle) and the healthy adjacent tissue (right). DEGs were calculated using FindMarkers function from Seurat with MAST as the method of choice. Genes with a *p*-value ≤ 0.05 and log2FoldChange ≥ 0.25 are displayed, visualized in volcano plots using GraphPad Prism 8. Several upregulated effector, (pre-)exhaused and tumor reactive genes are highlighted in orange.

**Figure 3 cancers-14-03164-f003:**
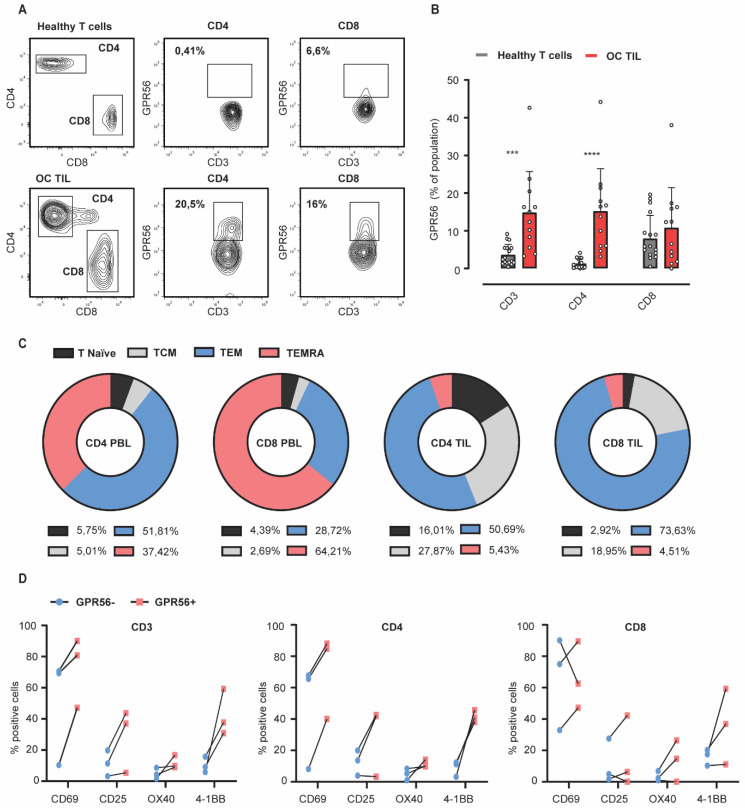
GPR56 expression is elevated on ovarian cancer TILs compared to healthy peripheral blood lymphocytes and associates with an effector memory phenotype. (**A**,**B**) GPR56 expression analyzed on CD3, CD4 and CD8 fractions of healthy T cells and ovarian cancer TILs (*n* > 10) by flow cytometry. A 2way ANOVA was performed to determine significance. (**C**) Different T cell developmental subsets (naïve, central memory, effector memory and effector memory RA (based on CD45RO and CCR7 expression) within the GPR56-positive CD4 and CD8 T cell fractions of peripheral blood lymphocytes and ovarian cancer TILs (*n* > 5). (**D**) Expression levels of effector (CD69 and CD25) and co-stimulatory (OX40 and 4-1BB) molecules within the GPR56-positive CD3, CD4 and CD8 T cell fractions of ovarian cancer TILs from 3 different donors. Individual donor variations between the GPR56+ and GPR56- fractions are displayed. Across the figure, *t*-test values: *** = *p* < 0.001, **** = *p* < 0.0001, all error bars in the figure represent SD.

**Figure 4 cancers-14-03164-f004:**
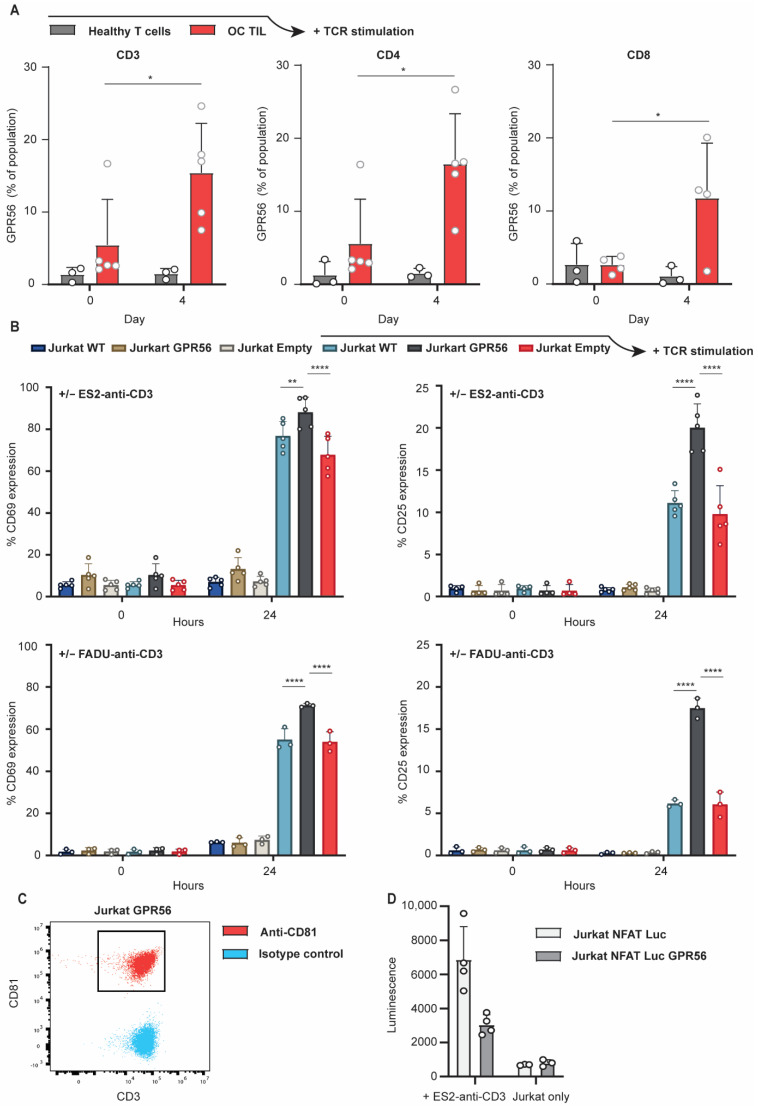
TCR stimulation upregulates GPR56 expression on TILs and enhances the activation status of T cells. (**A**) ES-2scFvCD3 and T cells (isolated from healthy human PBMC donors or from TILs, from digested ovarian cancer tumor samples) (Effector: Target ratio 10:1) co-cultured for 4 days. GPR56 expression analyzed within the CD3, CD4 and CD8 fractions at day 0 and day 3 of the co-culture. (**B**) ES-2scFvCD3 or FaDuscFvCD3 co-cultured with Jurkat WT, Jurkat GPR56 and Jurkat empty vector for 24 h. Expression levels of CD69 and CD25 were analyzed on the different Jurkat cell lines after 24 h of co-culture. 2way ANOVA’s were performed to determine significance. (**C**) CD81 expression on CD3-positive Jurkat GPR56 cells. (**D**) Luminescence from Jurkat NFAT Luc and Jurkat NFAT Luc GPR56 upon a 24-h co-culture with ES2-anti-CD3 at a 6:1 ratio or Jurkat cells alone (*n* = 4). Across the figure, *t*-test values: * = *p* < 0.05, ** = *p* < 0.01, **** = *p* < 0.0001, all error bars in the figure represent SD.

**Figure 5 cancers-14-03164-f005:**
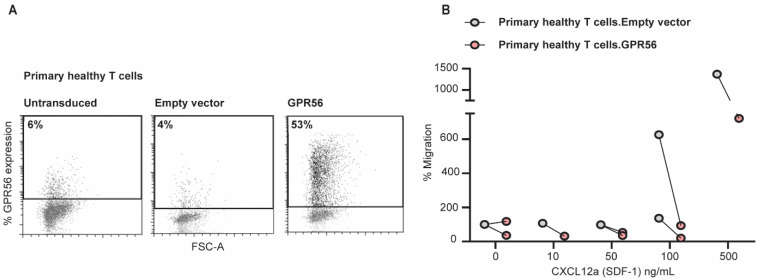
Ectopic GPR56 expression inhibits migration of primary healthy T cells. (**A**) Primary healthy T cells ectopically expressing GPR56 were generated by transducing T cells with lentivirus containing pRRL-SFFV-GPR56-iGFP or pRRL-SFFV-iGFP. After 3 days the GPR56 expression was evaluated by flow cytometry. (**B**) Migration in response to chemoattractant CXCL12a was performed with primary healthy T cells lentivirally transduced with pRRL-SFFV-GPR56-iGFP or an empty vector. Primary healthy T cells were loaded in a volume of 100 µL on transwell filters with a pore size of 5 µm. SDF-1 (CXCL12) was added as chemoattractant to the lower compartment at a concentration of 0–500 ng/mL in a total volume of 200 µL of migration buffer. After 2.5 h incubation at 37 °C, cells in the lower compartment were harvested and quantified by flow cytometry. The percentage of migration was calculated considering the migration of the primary healthy T cells transduced with the empty vector as 100%. A two-tailed unpaired t test was used to compare the levels of migration.

## Data Availability

The data that support the findings of this study are available from the corresponding author upon reasonable request.

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
