# Peer review of "The Novel Immune Checkpoint GPR56 Is Expressed on Tumor-Infiltrating Lymphocytes and Selectively Upregulated upon TCR Signaling"

_cancers, 2022, doi:10.3390/cancers14133164_

Round 1

Reviewer 1 Report

Martijn R. Vlaming et al. analyse GPR56 on different tumor samples and suggest GPR56 as a novel immune checkpoint exploitable to restoe T cell function. The topic is of great interest and among tumor types they also analyse ovarian cancer, a tumor that urgently need an efficient cure.

Follow some comments:

If to study different tumor type could be of some interest, especially in order to find a new therapeutical approach common for different tumor, on the other side, this way to procede could be superficial and a little bit unhelpful for the single tumor type.

1) It could have been better, for example to perform similar analyses with the same tumor samples. Is this still possible? For example is it possible to analyse different gene expression also on ovarian cancer samples?

2)Another important point is the analyses of cancer cells. If the authors want to suggest GPR56 as a new immune checkpoint useful for immunotherapeutic approach to cure cancer, they should check the presence of CD81 on lung/ovarian cancer cells.

Minor comments

1) It is necessary to know the gate strategy used to analysed T cells derived from tumor tissue. In addition it is necessary to clarify if the authors analyzed tissue resident or only all T cells derived from tumor tissue.

2) Regarding ovarian cancer, which tumor type has been analyzed?

3)Regarding tumors (lung, ovarian,...) it could be important to specify the precise anatomic region of the samples analysed.

4)In figure 2 axis label of dot plot are missed

Author Response

Dear Prof. Dr. Samuel C. Mok

Thank you for giving us the opportunity to submit a revised draft of our manuscript with the revised title “The novel immune checkpoint GPR56 is expressed on tumor-infiltrating lymphocytes and selectively upregulated upon TCR signaling” to Cancers. We appreciate the effort and the time that you and the reviewers have dedicated to provide valuable feedback on our manuscript and have carefully adapted the manuscript to address the reviewers comments.

Below please find a point-by-point response to the reviewers’ comments and concerns.

Comments from Reviewer 1

Major comments

  • Comment 1: If to study different tumor type could be of some interest, especially in order to find a new therapeutical approach common for different tumor, on the other side, this way to proceed could be superficial and a little bit unhelpful for the single tumor type.It could have been better, for example to perform similar analyses with the same tumor samples. Is this still possible? For example is it possible to analyse different gene expression also on ovarian cancer samples?

Response: Indeed, doing a paired RNAseq and functional assays would have been the best set-up. Unfortunately, at this moment we do not have patient material remaining from the exact primary samples examined in this study. Thus, we are unable to do this fully matched analysis. We would like to point out that the publicly available data set we used in this study comprised 500.000 immune cells from 217 patients, including patients with ovarian cancer.

  • Comment 2: Another important point is the analyses of cancer cells. If the authors want to suggest GPR56 as a new immune checkpoint useful for immunotherapeutic approach to cure cancer, they should check the presence of CD81 on lung/ovarian cancer cells.

Response: It is indeed of interest to know the expression of CD81, albeit with the caveat that CD81 has been functionally reported to be co-expressed with GP56 on NK cells (and not on cancer cells), where it regulates activity. Thus, CD81 expression should ideally be assessed on T cells. In the scRNA dataset, we confirmed CD81 expression in all tumor infiltrated T cell populations (see supplementary figure 2A). Further, in the Jurkat model system, expression of CD81 at the protein level was clearly detected (see Figure 4C of revised manuscript).

Minor comments

  • Comment 1: It is necessary to know the gate strategy used to analysed T cells derived from tumor tissue. In addition it is necessary to clarify if the authors analyzed tissue resident or only all T cells derived from tumor tissue.

Response: We have included the gating strategy followed in Supplementary Figure 1A. Further, we have evaluated the expression of GPR56 on CD103-positive and CD103-negative cells within the tumor infiltrated lymphocyte population, as it is considered one of the essential tissue resident markers (allowing distinction between stromally located and intra-tumorally located T cells). GPR56 was found expressed on both CD103-positive and CD103-negative cells (Supplementary figure 1B), with no significant differences between the two populations.

  • Comment 2: Regarding ovarian cancer, which tumor type has been analyzed?

Response: The analysis was done on high grade serous ovarian cancer.

  • Comment 3: Regarding tumors (lung, ovarian,...) it could be important to specify the precise anatomic region of the samples analysed.

Response: This would indeed be very interesting to know. However, it is impossible to retrieve this information as we receive the patient material anonymized from the pathology department with no access to surgical details.

Further, the samples used for the scRNAseq lung cancer TIL analysis: raw sequencing counts and metadata from the Gene Expression Omnibus (GEO) dataset GSE99254 was downloaded (T cell landscape of non-small cell lung cancer revealed by deep single-cell RNA sequencing), consisting of 12,346 T cell samples isolated from the blood, healthy adjacent tissue and tumor tissue of 14 treatment-naïve non-small-cell lung cancer (NSCLC) patients (Guo et al., 2018).

The other scRNAseq analysis in the current study is based on data from the tumor Immune Cell Atlas study (Nieto et al., 2020).

  • Comment 4: In figure 2 axis label of dot plot are missed

Response: In the revised manuscript, we have added it accordingly.

Reviewer 2 Report

In the present study, the authors indicate GPR56 as a possible checkpoint receptor of tumor infiltrating T cells. They first analyse RNA sequencing data sets from different tumor cell types and show that GPR56 is expressed in TILs, predominantly in CD8+ T cells with cytotoxic, tumor reactive, and exhausted phenotypes. Then, the authors analyse ovarian cancer patients and confirm the GPR56 protein expression in TILs. These latter experiments show that GPR56 is expressed both in CD4+ and CD8+ T cells. Comparative analysis of healthy donors PB, peritumoral, and tumoral tissues reveals that GPR56 is expressed, at lower extent, also in healthy conditions and marks different T cell subsets. Nevertheless, TCR stimulation can induce GPR56 up-regulation in TILs and not in healthy PB lymphocytes. Finally, the authors assess the possible functions of GPR56 on T cells by transfection experiments, which indicate that the expression of GPR56 associates with a reduced activation and migration capability of the cells.

The design of the study is well conceived and the experiments appear consistent. The manuscript is quite well written although some points may be improved. Some of the results do not properly correspond to the drawn conclusions.

Major points.

1) As also reported in the title, one of the messages of the study is that GPR56 is predominantly expressed in TILs. However, the authors show contrasting results on this point. Indeed, in NSCLC patients GPR56 expression does not appear to be predominantly expressed in TILs, rather it is slightly higher in blood and adjacent healthy tissues (Figure 2C). The authors should comment on this and amend the title accordingly.

2) The authors indicate the GPR56+ TILs as tumour reactive, exhausted T cells. Actually, the shown data do not unambiguously indicate an exhausted phenotype for these cells. Compared to PB GPR56+ T cells, GPR56+ TILs show higher expression of IFNg, Ki67 and granzymes (Figure 2D), which, on the whole, do not mark exhausted cells. Therefore, the definition of the phenotype of GPR56+ TILs should be reconsidered and discussed. It may be appropriate to discuss the definition of cytotoxic/exhausted/tumor reactive signature, as these may represent contrasting features.

3) The composition in terms of TEM, TCM, and TEMRA of PB and TIL T cells should be evaluated and compared, since the distribution of GPR56+ cells may simply reflect the proportion of the different T cell subsets in healthy PBL and TIL.

Minor.

a) Lines 239-250. The sentences describing the comparative analyses of the GPR56+ cells in the different cell sources (tumour, healthy peri-tumour, and blood) may be modified for clarity.

b) Figure 2A. In each panel, the colour represents the expression intensity of the indicated genes (CD3, CD4, etc…) according to the colour key under figure 2A. If this is the case, the label GPR56 on the colour key is not correct. Isn’t it? The legend also should be modified accordingly.

c) lines 261 and 314. “pre-dominantly” should be replaced by “predominantly”.

d) Lines 314-317. This sentence may be modified for clarity.

e) Line 339. Which “current study” are the authors referring to?

f) The correct term for T cells undergoing exhaustion is “exhausted” and not “exhaustive”.

Author Response

Dear Prof. Dr. Samuel C. Mok

Thank you for giving us the opportunity to submit a revised draft of our manuscript with the revised title “The novel immune checkpoint GPR56 is expressed on tumor-infiltrating lymphocytes and selectively upregulated upon TCR signaling” to Cancers. We appreciate the effort and the time that you and the reviewers have dedicated to provide valuable feedback on our manuscript and have carefully adapted the manuscript to address the reviewers comments.

Below please find a point-by-point response to the reviewers’ comments and concerns.

Comments from Reviewer 2

Major comments

  • Comment 1: As also reported in the title, one of the messages of the study is that GPR56 is predominantly expressed in TILs. However, the authors show contrasting results on this point. Indeed, in NSCLC patients GPR56 expression does not appear to be predominantly expressed in TILs, rather it is slightly higher in blood and adjacent healthy tissues (Figure 2C). The authors should comment on this and amend the title accordingly.

Response: We agree with the reviewer’s comment and have changed the title to more accurately reflect the message of our manuscript to “The novel immune checkpoint GPR56 is expressed on tumor-infiltrating lymphocytes and selectively upregulated upon TCR signaling”.

Indeed, for NSCLC patients GPR56 was detected at all anatomical isolation sites, with expression levels being slightly higher within the blood and healthy adjacent tissue fractions compared to the expression levels found in cells isolated from tumor tissue (Figure 2C). Importantly, the GPR56-positive TILs displayed an exhausted and tumor-reactive phenotype, which also fits with the findings in epithelial ovarian cancer TILs and the functional Jurkat activity data added to the revised manuscript in Figure 4D.

  • Comment 2: The authors indicate the GPR56+ TILs as tumour reactive, exhausted T cells. Actually, the shown data do not unambiguously indicate an exhausted phenotype for these cells. Compared to PB GPR56+ T cells, GPR56+ TILs show higher expression of IFNg, Ki67 and granzymes (Figure 2D), which, on the whole, do not mark exhausted cells. Therefore, the definition of the phenotype of GPR56+ TILs should be reconsidered and discussed. It may be appropriate to discuss the definition of cytotoxic/exhausted/tumor reactive signature, as these may represent contrasting features.

Response: Indeed, the statement that these cells are exhausted might be a bit strong based on the provided evidence. We suggest changing this to a (pre-)exhausted phenotype, encompassing both, which would be consistent with the higher expression of IFNg, Ki67 and granzymes.

  • Comment 3: The composition in terms of TEM, TCM, and TEMRA of PB and TIL T cells should be evaluated and compared, since the distribution of GPR56+ cells may simply reflect the proportion of the different T cell subsets in healthy PBL and TIL.

Response: We analyzed the expression within these various differentiation stages, as illustrated in piecharts in Figure 3C of the original and revised manuscript. In brief, GPR56-expressing TILs were predominantly identified within the effector memory (TEM) and central memory (TCM) T cell subsets, both on CD4 and CD8 TILs. In the PB, GPR56 expression was limited to effector memory (TEM) and terminally differentiated T cells (TEMRA).

Minor comments

  • Comment 1: Lines 239-250. The sentences describing the comparative analyses of the GPR56+ cells in the different cell sources (tumour, healthy peri-tumour, and blood) may be modified for clarity.

Response: We have modified the text accordingly.

  • Comment 2: Figure 2A. In each panel, the colour represents the expression intensity of the indicated genes (CD3, CD4, etc…) according to the colour key under figure 2A. If this is the case, the label GPR56 on the colour key is not correct. Isn’t it? The legend also should be modified accordingly.

Response: We have modified it accordingly.

  • Comment 3: lines 261 and 314. “pre-dominantly” should be replaced by “predominantly”.

Response: We have modified it accordingly.

  • Comment 4: Lines 314-317. This sentence may be modified for clarity.

Response: We have modified it accordingly.

  • Comment 5: Line 339. Which “current study” are the authors referring to?

Response: We are referring to the present manuscript.

  • Comment 6: The correct term for T cells undergoing exhaustion is “exhausted” and not “exhaustive”.

Response: We have modified it accordingly.

Reviewer 3 Report

In the manuscript “The Novel Immune Checkpoint GPR56 Is Predominantly Expressed and Selectively Upregulated in Tumor-Infiltrating and Tumor-Reactive Lymphocytes”, Martijn R. Vlaming et al. reported that GPR56 expression was detected predominantly in tumor-infiltrating CD8 T cells with a cytotoxic and exhaustive phenotype and GPR56 is a potential immune-checkpoint. The most attractive part of this manuscript is “novel Immune checkpoint GPR56”. While the topic is fascinating, there are some limitations in this manuscript.

1.       The major limitation is that the scRNA-seq data analysis was premised on the hypothesis “GPR56 is an immune checkpoint”. The authors analyzed the level of GPR56 in different cell subpopulations and different gene expressions in GPR56+ vs. GPR56- cells. However, many genes can be upregulated in the exhausted T cells, but not only a few of them can be considered immune checkpoints.  If GPR56 is not an immune checkpoint, the analysis here will be meaningless. However, this manuscript has minimal data to support that GPR56 is an immune checkpoint in T cells. To prove the hypothesis, the authors should provide pieces of evidence of 1) expression of GPR56 on exhausted T cells (some data); 2) inhibitory role of GPR56 in T cell function (no anti-tumor function); and 3) blockade of GPR56 can restore T-cell function (no data).

2.       Line 199-200 and figure 1B. It shows that Recently activated CD4 T cells have the highest level of GPR56, followed by Naive-memory CD4 T cells, other than cytotoxic CD8 T cells and terminally exhausted CD8 T cells. The mean of the expression level should be shown in figure 1B so that it would be easier to compare.

3.       Regression analysis should be done to find the correction between GPR56 with other genes instead of comparing GPR56+ vs. GPR56- populations in figures 1C and 2D.

4.       Labels should be shown in the flow cytometry plots in Figure 3A.

5.       In vitro migration assay data is weak evidence for T cell function. Anti-CD3 scFv expressing tumor cells can be used as the target cell to assess T cell cytotoxicity. T cell-mediated killing assays should be performed using the ES2-anti-CD3 and FADU-anti-CD3 cell lines. In this way, it is possible to compare the cytotoxic activity of T cells with/without GPR56 (WT T cell vs. GPR56-transduced T cell; WT T cell vs. GPR56 KO T cells).

Author Response

Dear Prof. Dr. Samuel C. Mok

Thank you for giving us the opportunity to submit a revised draft of our manuscript with the revised title “The novel immune checkpoint GPR56 is expressed on tumor-infiltrating lymphocytes and selectively upregulated upon TCR signaling” to Cancers. We appreciate the effort and the time that you and the reviewers have dedicated to provide valuable feedback on our manuscript and have carefully adapted the manuscript to address the reviewers comments.

Below please find a point-by-point response to the reviewers’ comments and concerns. 

Comments from Reviewer 3

  • Comment 1: The major limitation is that the scRNA-seq data analysis was premised on the hypothesis “GPR56 is an immune checkpoint”. The authors analyzed the level of GPR56 in different cell subpopulations and different gene expressions in GPR56+ vs. GPR56- cells. However, many genes can be upregulated in the exhausted T cells, but not only a few of them can be considered immune checkpoints.  If GPR56 is not an immune checkpoint, the analysis here will be meaningless. However, this manuscript has minimal data to support that GPR56 is an immune checkpoint in T cells. To prove the hypothesis, the authors should provide pieces of evidence of 1) expression of GPR56 on exhausted T cells (some data); 2) inhibitory role of GPR56 in T cell function (no anti-tumor function); and 3) blockade of GPR56 can restore T-cell function (no data).

Response: To substantiate the hypothesis that GPR56 is a checkpoint on T cells, we ectopically expressed GPR56 on NFAT Reporter-luciferase-Jurkat (Jurkat.NFAT-luc). In Jurkat.NFAT-luc the firefly luciferase gene is under the control of the NFAT response element and luminescence is a direct measure of T cell activation. Co-culture of GPR56-expressing Jurkat.NFAT-luc cells with ES2-anti-CD3 cells for 24 hours activated both GPR56-expressing and  control Jurkat.NFAT-luc, as evidenced by increased luminescence compared to Jurkat.NFAT-luc alone (Figure 4D). Importantly, luminescence by GPR56-expressing Jurkat.NFAT-luc cells was significantly reduced compared to control Jurkat.NFAT-luc cells (Figure 4D, left, light vs dark grey). Thus, ectopic expression of GPR56 directly inhibited T cell activation.

Note that blockade of GPR56 with an antibody is of great interest, but at the moment not feasible. Specifically, the partner of GPR56 on T cells has yet to be validated, with CD81 being demonstrated on NK cells but not (yet) on T cells. Therefore, selection of appropriate antagonistic antibody is only possible upon further mechanistic insight on GPR56 biology on T cells.

  • Comment 2: Line 199-200 and figure 1B. It shows that Recently activated CD4 T cells have the highest level of GPR56, followed by Naive-memory CD4 T cells, other than cytotoxic CD8 T cells and terminally exhausted CD8 T cells. The mean of the expression level should be shown in figure 1B so that it would be easier to compare.

Response: We have modified the text accordingly. Additionally, we incorporated a violin plot with mean expression values in the supplementary figure 2B.

  • Comment 3: Regression analysis should be done to find the correction between GPR56 with other genes instead of comparing GPR56+ vs. GPR56- populations in figures 1C and 2D.

Response: Thank you for your suggestion.  We have incorporated the correlation analysis into the supplementary figure 3A and 3B. We used ggcorr function in R to do pair wise Pearson correlation coefficient calculation for different genes and plot the data in the form of correlation matrix.

  • Comment 4: Labels should be shown in the flow cytometry plots in Figure 3A.

Response: We have modified it accordingly.

  • Comment 5: In vitro migration assay data is weak evidence for T cell function. Anti-CD3 scFv expressing tumor cells can be used as the target cell to assess T cell cytotoxicity. T cell-mediated killing assays should be performed using the ES2-anti-CD3 and FADU-anti-CD3 cell lines. In this way, it is possible to compare the cytotoxic activity of T cells with/without GPR56 (WT T cell vs. GPR56-transduced T cell; WT T cell vs. GPR56 KO T cells).

Response:

Next to the activation marker expression data and the in vitro migration assay, we added T cell activation data based on the GPR56-expressing Jurkat.NFAT-luc cell lines (as described in response to comment 1). Using this cell line, we were able to confirm an inhibitory effect of GPR56 on T cell function. Considering the timeframe of revision, experiments to assess the effect on GPR56 on anti-tumor activity, and the possible blockade by a GPR56 antagonist, will be reported on in follow-up studies. 

Round 2

Reviewer 2 Report

The new version of the manuscript still requires attention from the authors.

The revised manuscript does not include some of the changes indicated in the authors’ replies.

Specifically:

a) The term “pre-exhausted” phenotype has not been inserted. Neither it has been commented the presence of activation markers together with exhaustion markers in TIL. This point requires to be discussed in the “Discussion” section.

b) Lines 239-250 (original version of the manuscript) have not been modified.

Author Response

Comments from Reviewer 2

Minor comments

  • Comment 1: The term “pre-exhausted” phenotype has not been inserted. Neither it has been commented the presence of activation markers together with exhaustion markers in TIL. This point requires to be discussed in the “Discussion” section.

Response: We have incorporated the term “pre-exhausted” into the revised manuscript and adapted the texts accordingly.

  • Comment 2: Lines 239-250 (original version of the manuscript) have not been modified.

Response: This is line 251-261 in the revised manuscript. We have modified the text accordingly to make it clearer.

Reviewer 3 Report

This manuscript version has been improved a lot, although killing assays are still strongly recommended to assess the roles of GPR56 in T cells.  Labels in supplementary figure 3 are too small to read, which should be improved. 

Author Response

Comments from Reviewer 3

  • Comment 1: This manuscript version has been improved a lot, although killing assays are still strongly recommended to assess the roles of GPR56 in T cells.  Labels in supplementary figure 3 are too small to read, which should be improved.

Response: Thank you for your suggestion. We have modified it accordingly.